# Can smart policies solve the sand mining problem?

Michael Hübler[1,2☯¤a]*, Frank Pothen[1☯¤b]

**1** Institute for Environmental Economics and World Trade, Leibniz University Hannover, Hannover, Germany, **2** Center for International Development and Environmental Research ZEU, Justus Liebig University Giessen, Giessen, Germany

☯ These authors contributed equally to this work.
¤a Current address: Agricultural, Food and Environmental Policy, Institute for Agricultural Policy and Market Research, Justus Liebig University Giessen, Giessen, Germany
¤b Current address: Fraunhofer Center for International Management and Knowledge Economy IMW, Center for Economics of Materials CEM, Halle (Saale), Germany
* michael.huebler@agrar.uni-giessen.de

## Abstract

While sand has become a scarce essential resource for construction and land reclamation worldwide, its extraction causes severe ecological damage and high social costs. To derive policy solutions to this paramount global challenge with broad applicability, this model-based analysis exemplarily studies sand trade from Southeast Asia to Singapore. Accordingly, a coordinated transboundary sand output tax reduces sand mining to a large extent, while the economic costs are small for the sand importer and slightly positive for the exporters. As a novel policy implementation approach, a "Sand Extraction Allowances Trading Scheme" is proposed, which helps sustainably balance the importer's economic growth with the exporters' economic development.

## 1. Introduction

This article increases public awareness of a paramount global challenge which is hardly known and rarely discussed in the mainstream media and the policy debate: On the one hand, the dependence of the world economy on sand used for any kind of building and infrastructure. On the other hand, the scarcity of usable sand and the vast ecological and social costs of sand mining. To this end, this article provides the first economic analysis of sand mining, sand trade and related public policies. For this purpose, it presents the first advanced trade model, and in the broader sense, probably the first general equilibrium model that incorporates sand extraction and trade explicitly and disaggregates the data accordingly.

While huge amounts of sand are essential for construction (of buildings and infrastructure), for land reclamation (due to land scarcity in growing urban areas) and for shoreline restoration (because of coastal erosion) worldwide, it is often overlooked that sand has become scarce and that sand extraction causes severe ecological damage in oceans, in rivers (and deltas) and on beaches. Although sand mining has been steadily enhanced by globalization and economic growth, economists and political scientists have rarely addressed this topic and provide no

**Data Availability Statement:** The data underlying this study are not owned by the authors. The GTAP data can be purchased at https://www.gtap.agecon.purdue.edu/ and the UN Comtrade data can be obtained from https://comtrade.un.org/. The study findings can be replicated in their entirety by

directly obtaining the data from these sources and following the modelling procedure described in our technical appendix. The authors confirm that they did not special access to the data that other researchers would not have.

**Funding:** This study was supported by the German Federal Ministry of Education and Research in the form of a grant (BMBF, project ROCHADE, Grant No. 01LA1828C) awarded to M. H. and by the Federal State of Saxony-Anhalt and European Regional Development Fund (ERDF) in the form of a grant (Grant No. ZS/2018/12/96265 (Economic Structural Dynamics)) awarded to FP. The funders had no role in study design, data collection and analysis, decision to publish, or preparation of the manuscript.

**Competing interests:** The authors have declared that no competing interests exist.

solutions. Thus, drawing on insights from various disciplines, this article discusses output taxes, or equivalently cap and trade policies, compared with import and export tariffs, regulating sand extraction and trade. Although such policies have been implemented to reduce $CO_2$ and $SO_2$ emissions, their application to resource extraction is novel. (A comparable tax on sand extraction and imports currently exists in the United Kingdom, https://www.gov.uk/green-taxes-and-reliefs/aggregates-levy, accessed 06/2018). The introduction of these policies follows the UNEP [1]:

*"We need to reconcile relevant global policies and standards with local sand availability, development imperatives and standards and enforcement realities. We need to recognize the interdependence between countries and sectors and learn lessons on how to manage this critical resource sustainably"* [1].

As a prominent example, this article assesses policies regulating sand exports from developing countries in Southeast Asia to Singapore [2]; a topic that seems to have received no attention from economists. The procedure of this exemplary policy assessment and the (qualitative) policy recommendations can directly be transferred to other countries and expanding (mega) cities in the world, especially to (emerging) municipalities extending or improving their infrastructure and housing.

Due to its impressive socioeconomic development, Singapore has become one of the richest countries in the world and Southeast Asia's front-runner economy. Singapore's government fosters the country's front-runner position by promoting high social and environmental standards. Nonetheless, a crucial challenge has existed for decades: The amount of Singapore's available land is limited, which has resulted in skyrocketing housing and infrastructure construction as well as massive land reclamation. These measures require vast amounts of sand and gravel. (Throughout the article, the term "sand" comprises both sand and gravel.) As Singapore's own sand deposits have been exploited a long time ago, it has become the world's top sand importer [3, 4] by purchasing sand from neighboring developing countries in Southeast Asia–with substantial negative externalities imposed on these countries.

*Although sand extraction creates revenues for exporters, its distributional and ecological effects on the exporting developing countries are often disastrous (based on the interdisciplinary literature summarized in Section 2).*

While the mining companies earn profits, the local population, ecosystems and biodiversity suffer from the destruction of the Mekong, other rivers, beaches and small islands in the South China Sea caused by sand mining. In the absence of technological or economic alternatives, Singapore's economic growth inevitably depends on sand imports, despite sand export bans. Driven by Singapore's ongoing growth and plans for more intensive land use and further land reclamation, this dilemma will likely be exacerbated in the future.

Sand extraction, however, is a paramount global challenge. According to the "Precautionary Principle" of the Rio Declaration, research into the economic, political, geographic and physical facets of sand mining is urgently required [5].

*Sand and gravel represent the most important solid extracted material in the world with an extraction rate by far exceeding the renewal rate* [3, 4].

In addition to the essential importance to construction, sand is increasingly used to restore or extend shorelines while the ocean continuously erodes them. Unfortunately, natural sand

supply by rivers to the oceans is increasingly hindered by dams, and the abundantly available sand from deserts is not suitable for construction. Thus, due to lack of suitable sand deposits, beaches (e.g., in North Africa or Australia) [6], river and ocean beds are destroyed by legal and illegal sand extractors [4].

Consequently, the policy solutions discussed in this article are of broad international relevance and can be transferred to other countries and (mega) cities. The results are not only highly relevant for the sand and gravel sector, but also for the construction sector and countries' entire urban and infrastructural economic development.

The article is structured as follows. While Section 2 summarizes the natural science and interdisciplinary literature on sand mining, Section 3 relates this article to the couple of working papers in the policy-economy domain. Section 4 discusses policy options to mitigate the detrimental effects of sand mining, Section 5 summarizes the advanced trade model, and Section 6 defines the policy scenarios under scrutiny. While Section 7 presents the policy simulation results, Section 8 discusses model limitations and parameter uncertainty. Based on this, Section 9 derives policy solutions. Section 10 concludes.

## 2. Ecological damages caused by sand mining

This section summarizes the current state of the research into the ecological and social effects of sand mining within different scientific disciplines, beginning with general insights, followed by evidence from Southeast Asia. The results of this research are the basis for the following economic policy analysis.

Sand mining causes various detrimental environmental and socioeconomic effects. It harms vulnerable habitats and protected areas such as mangroves, seagrass beds and coral reefs and negatively affects the biodiversity in these habitats [4, 5]. The conservation of such unique habitats and areas requires the prohibition of mining activities as well as strict local control and enforcement. The extension of prohibitive regulation to *all* areas beyond protected areas, however, is incompatible with rising sand demand. Nonetheless, detrimental effects of sand mining also occur outside protected areas [4, 5, 7–9]: water flows and marine currents can be changed; the benthic fauna is destroyed; fish and crab populations decline in rivers and the sea causing a loss of fishermen's livelihood; the pressure on endangered species increases with a negative impact on biodiversity; bridges, river embankments and coastal infrastructure can be damaged; flood regulation and protection are impeded, and the agricultural use of floodplains can be restrained; the water table can be lowered and water supply impeded; and recreational functions can be reduced.

Recent geographic studies of Southeast Asia's Mekong River and its delta provide evidence of geomorphic changes such as riverbed incision, subsidence and coastal erosion [10–12]. Further geographic studies identify sand mining as a major contributor to the geomorphic changes [13, 14]. Similarly, there is evidence that maritime sand mining caused the decline of small islands in the South China Sea and Indonesia [4, 5].

## 3. Economics literature on sand mining

Against the backdrop of the studies summarized in the following section, our article provides the first economic analysis of sand mining, sand trade and suitable policies to mitigate the negative effects of sand mining for society and the environment. To this end, it incorporates sand extraction and trade explicitly and disaggregates the data accordingly. To our knowledge, it provides the first advanced trade model, or in the broader sense, general equilibrium model representing sand in such an explicit way.

Despite the recently growing number of sand mining studies in other scientific disciplines, the economic analysis of sand mining and trade is, to our knowledge, limited to two working papers. Hoogmartens et al. [15] examine sand extraction in Flanders based on a one-sector *Hotelling*-type growth model with resource extraction. The researchers estimate that the taxation of construction and sand extraction would extend the time frame until sand's depletion from 30 to over 40 years.

Franke [16] presents a narrative based on the *world-systems theory* to discuss the core-periphery structure of Singapore and its surrounding Southeast Asian countries focusing on sand trade. The author argues that Singapore has grown at the expense of its periphery while export bans in Southeast Asian countries have been undermined by illegal sand trade.

## 4. Policy options to reduce sand mining

This section discusses relevant policy options to address the sand mining problem. Based on this discussion, the model analysis presented in the following sections will examine a cost-efficient implementation of given targets for the allowed extent of sand mining (outside protected areas).

To this end, this article will evaluate various market-based policy instruments imposing a tax or, equivalently, a price emerging from a S*and Extraction Allowances Trading Scheme (SEATS)*, on sand output (sales of extracted sand), or imposing a tax on sand trade (via import or export tariffs) in addition to the market price of sand. In practice (not visible in the model), these instruments can be combined with a *certificate* guaranteeing that the traded sand was extracted outside *protected areas*. The increased gross price of sand is expected to reduce sand demand and trade while making alternative materials (wood, steel, recycled concrete or other recycled materials, byproducts, prepared sand from deserts, new mixtures etc.) and construction techniques using these materials more attractive compared to standard concrete building techniques [4]. (In the model, these substitution effects are represented in an abstract, aggregate way without an explicit representation of specific materials, techniques or research and development.)

Leaving aside, in this analysis, (changes in) the *damages* caused by sand mining, the imposed policy instruments create different welfare and distributional effects for consumers and producers in the exporting countries and in Singapore, the importing country. Hence, we search for a suitable policy solution that keeps welfare losses of exporters and the importer low and reduces the discrepancy between winners and losers of sand mining and the imposed policy. The currently and previously implemented export bans do not generate official revenues but instead encourage smuggling and unofficial rent-seeking. The introduction of taxes or tariffs, on the contrary, would generate official state revenues that can be used for social or environmental purposes related to sand mining, particularly as a compensation to those whose welfare is reduced by sand mining.

In theory, a site-specific *Pigou* tax that internalizes the social (ecological) costs of each sand miner with a tax rate equal to the social marginal damage would be optimal. Damages are, however, unlikely to be restricted to the local surroundings of an extraction site. Instead, they affect the entire downstream river system or the coastal area around the extraction site. Thus, following the *Samuelson* rule, the socially optimal tax rate should reflect the sum of *all* marginal damages for which the producer is responsible. Because the Mekong flows through several countries, damages are transboundary, i.e., the creators of and the sufferers from the externality are not located within the same jurisdiction. This complicates the policy implementation. Indeed, history has shown that uncoordinated regulation of sand mining by single Southeast Asian governments results in a shift of sand mining to other countries (cf. the

supplementary part I, S1 Fig D1 in S1 Appendix). Thus, a harmonized transboundary policy solution with a uniform tax rate including all relevant exporters and importers is recommendable as a practical solution (cf. the call for an international framework) [4] and will be used in this analysis. As in the climate policy case, this approach implies that the same total transboundary marginal damage is attributed to all Southeast Asian sand miners, resulting in an economically efficient solution. (In the model, there is one representative sand miner, i.e., one sand sector, per region engaged in sand trade.)

## 5. The advanced trade model

This section summarizes the advanced trade model used for the policy analysis. Details can be found in the S1 Appendix.

For the economic policy analysis, we set up a global *Eaton and Kortum* type [17] general equilibrium model of international trade with Ricardian specialization similar to that of Caliendo and Parro [18]. It uses nested constant elasticity of substitution (CES) production and consumption functions (see S1 Figs A1 and A2 and S1 Table A3 in the supplementary part II in S1 Appendix) similar to Pothen and Hübler [19]. This model type enables the explicit theory-based representation of international trade (with the sectoral trade elasticities shown in S1 Table A4 in S1 Appendix).

In contrast to Pothen and Hübler [19], the model equations are expressed as changes between a counterfactual and the baseline as suggested by Dekle et al. [20]. Trade flows are determined by international differentials of fundamental productivity and input costs as well as (iceberg) trade costs. Unlike *Armington* trade models, the *Eaton and Kortum* model assumes neither home bias nor regional preferences. Instead, Ricardian specialization creates productivity gains via the endogenous choice of the lowest-cost varieties of each good (sector). Existing sand trade policies (export bans) are eliminated from the benchmark data based on an econometric estimation before imposing the policies being analyzed (see Section 4.4 of part II Appendix). The model is formulated as a *Mixed Complementarity Problem (MCP)* with market clearance, zero-profit and income balance conditions as well as specific equations defining international trade and sand policies. The supplementary part II S1 Appendix provides more information about the model, including mathematical details in Section 5.

The aggregation of countries and sectors is based on the GTAP 9 dataset for the benchmark year 2011 as well as the UN Comtrade [3] data. (The Global Trade Analysis Project data can be purchased at https://www.gtap.agecon.purdue.edu/default.asp. Additionally, 16 countries/ regions of the world and corresponding trade connections, except sand, are represented by the model but not specifically analyzed, see S1 Table A1 in the S1 Appendix. The model incorporates 15 production sectors and the investment good sector and the corresponding input-output connections with each other and with final consumption, see S1 Table A2 in the S1 Appendix. There are two production factors, capital including natural resources and labor, with fixed region-specific endowments.) S1 Fig D1 in the supplementary part I in S1 Appendix illustrates Singapore's sand imports according to official statistics. Because sand is not explicitly represented in the available global datasets, we disaggregate sand extraction from the remaining mining sector in the exporter countries. Similarly, we disaggregate sand as an input to Singapore's production sectors and (private and public) consumption. The disaggregation procedure and the calibration to data are explained in Sections 4.2 to 4.4 of the supplementary part II S1 Appendix.

## 6. Sand policy scenarios

Against this backdrop, an analysis of policy options to reduce sand extraction (outside protected areas) is required. For this purpose, this section defines and characterizes the policy scenarios under scrutiny.

As a case study, we focus on the sand trade of five Southeast Asian exporters (ordered by export volumes), *Cambodia (KHM)*, *Myanmar (MMR)*, *the Philippines (PHL)*, *Malaysia (MYS)* and *Vietnam (VNM)*, with the importer, *Singapore (SGP)*. (Instead of being exported to Singapore, the extracted sand can be sold in the same country or exported to the remaining Southeast Asian exporters, *KHM*, *MYS*, *MMR*, *PHL* and *VNM*; see S1 Table A5 in S1 Appendix. Sand exports to other countries and sand imports from other countries to Singapore, *SGP*, are negligible and are left out.) The scrutinized policy instruments and the qualitative results are directly applicable to other countries, urban areas or (mega) cities too, given that sand scarcity is a global phenomenon.

Because the Mekong flows through several Southeast Asian countries, damages are transboundary, which complicates the policy implementation. Previous policy approaches have shown that uncoordinated regulation of sand mining by single Southeast Asian governments results in a shift of sand mining to other countries (see the supplementary part I, S1 Fig D1 in S1 Appendix). Thus, a harmonized transboundary policy solution with a uniform tax rate including all relevant exporters and importers will be analyzed (cf. the call for an international framework) [4].

All types of taxation analyzed in the following policy scenarios implement a sand tax at a rate that reflects the actual physical sand content such that the damages created by sand extraction are proportional to the amount of extracted sand (for implementation details, see Section 4.5 of the Appendix). The sand tax can be interpreted as the market price emerging from a S*and Extraction Allowances Trading Scheme (SEATS)* with a fixed total amount of extracted sand (the "cap and trade" scheme). Although emissions trading schemes have been successful in tackling $CO_2$ and $SO_2$ emissions [21, 22], the application to resource extraction is, to our knowledge, novel.

Leaving aside, in this analysis, (changes in) the valuation of the damages caused by sand mining, the imposed policy instruments create different welfare and distributional effects for sand consumers and producers. Hence, we search for an efficient, fair and implementable policy solution that keeps the resulting welfare losses low and reduces the discrepancy between winners and losers.

In summary, we consider three market-based policy instruments in three policy scenarios:

1. *OutTax* implements a sand output (extraction) tax. The tax revenues accrue to the budget of the corresponding country as a lump-sum. This tax can be equivalently implemented as a cap and trade policy *(Sand Extraction Allowances Trading Scheme, SEATS)*.

2. *ExpTax* implements a sand export tax (tariff) imposed by the Southeast Asian exporting countries. In contrast to the OutTax scenario, only the exported part of extracted sand is taxed and generates revenues for the exporting country.

3. *ImpTax* implements an import tax (tariff) that Singapore's government imposes on sand imports from Southeast Asia. The tax revenues accrue to the importer, here Singapore.

These three policy scenarios can be characterized as follows. The first policy scenario denoted by OutTax, examines a sand output tax suggested by [4, 7, 15]. The tax is imposed on the sales of extracted sand. The tax rates are internationally harmonized across exporters. In this scenario, the tax is imposed on the total sand extraction, regardless of whether the sand is sold in the domestic market or exported to Singapore. The tax revenues accrue to the representative consumer of the corresponding country as a lump-sum.

The second policy scenario denoted by ExpTax, studies a uniform sand export tax (tariff) imposed by the Southeast Asian exporting countries. Similar to the OutTax scenario, this scenario requires an internationally coordinated policy solution. In contrast to the tax scenario,

only the exported part of extracted sand is taxed and generates revenues for the representative consumer of the exporting country. The tax rate is internationally harmonized and reflects the actual sand content, as argued above. In this respect, this policy mimics the *border carbon adjustments* that have been studied in depth in the climate policy domain [23] and discussed controversially regarding the compliance with the World Trade Organization legislation.

The third policy scenario denoted by ImpTax, analyzes an *import tax* (tariff) that Singapore's government imposes on sand imports from Southeast Asia. As before, the tax rate reflects the actual sand content. This policy scenario has the advantage that the Singaporean government can as a single actor decide on, implement and administer the import tariff in a unified way for all sand exporters without bargaining for an internationally coordinated solution. Unlike the previous two scenarios, the tariff revenues accrue to the Singaporean consumer (via the Singaporean government).

To evaluate the policy effectiveness, we vary the stringency of each instrument (i.e., the tax rate) and observe the impact on the reduction of sand extraction and welfare of Singapore and the Southeast Asian sand exporters based on data for the benchmark year 2011.

## 7. Policy scenario simulation results

This section presents, summarizes and interprets the simulation results. The supplementary material contains a number of figures presenting further results. S1 Fig D1 of the supplementary part I in S1 Appendix illustrates the quantity-based export shares of Singapore's trading partners in the model's benchmark year 2011 and their variation across other years according to the official UN Comtrade [3] statistics. In 2011, Cambodia was the largest exporter (with a share of 62%), followed by Myanmar (28%). In what follows, let us refer to them as the major exporters. On the contrary, let us refer to the following countries as the minor exporters: the Philippines (9%), followed by Malaysia (1%) and Vietnam (close to 0%). Indonesia and other sand suppliers are left out because of negligible sand supply in all inspected years.

Figs 1 and 2 show the results of the model-based policy scenario simulations. (The S1 Fig 1 in S1 Appendix replicates Figs 1 and 2 of the main text.) Fig 1 displays the sand tax rate (marginal abatement costs) under scenario OutTax over the corresponding reduction of total sand extraction in all considered exporting countries. (To obtain this relationship, either the sand tax rate or the sand (reduction) amount (*SEATS*) is fixed, while the other one emerges endogenously. The total sand extraction is calculated as the sum of extracted quantities in all extracting countries.) An output tax rate of US-$10 (US-$20) per (metric) ton of sand achieves a reduction of almost 60% (respectively, 70%).

Fig 2 displays the corresponding graphs under ImpTax and ExpTax. Skyrocketing tax rates of US-$100 (US-$700) per ton of sand induce only a 1.5 to 2% (respectively, 2.5 to 3%) reduction. There are two explanations. First, taxation of traded sand induces a shift from exports to domestic sales in the exporting countries and, in the case of the import tax, a shift to other importers. Second, especially in Malaysia, the Philippines and Vietnam, the benchmark data contain substantial amounts of domestic sand sales. Trade policy, however, is not able to reduce domestic sales. The export tax (ExpTax) achieves a slightly smaller reduction of sand extraction than that achieved by the import tax (ImpTax) because the former generates revenues within the exporting countries that are to some extent spent on domestically extracted sand. (The import tax does not create this effect in Singapore because Singapore does not possess domestic sand reserves.)

The supplementary S1 Figs 2 and 3 in S1 Appendix show country-specific relative welfare effects (where gains are positive) as a function of the corresponding reduction of total sand extraction. The major exporters can achieve welfare gains by imposing a uniform export tax

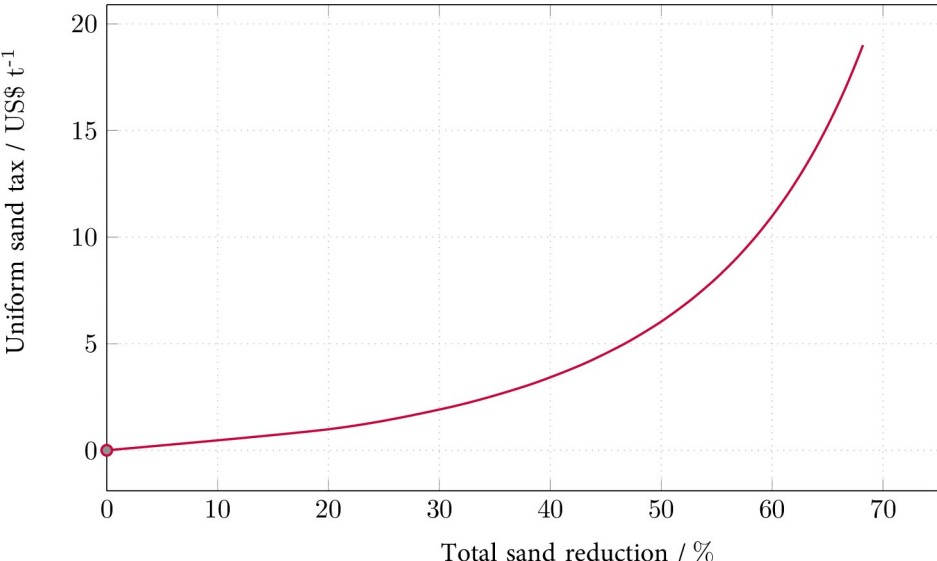

**Fig 1. Tax rate (marginal reduction cost) as a function of the total reduction of sand extraction under the output tax (OutTax).**

on sand in all exporting countries (ExpTax), whereas the minor exporters lose. With a total sand extraction reduction of 2.6%, the largest exporter Cambodia achieves the maximum welfare gain of 60%, and Myanmar achieves ca. 1.4% gain compared to the benchmark scenario of having no sand policy, notably, at an unrealistic tax rate of approximately US-$800 per ton. If the export tax is replaced by the Singaporean import tax (ImpTax), all exporters will become worse off than without any sand policy because the tax revenues accrue to Singapore. The welfare effects of the import tax are, however, small (far below 1%). Singapore, on the contrary, gains almost 3% from the import tax but loses almost 3% due to the export tax, compared to the benchmark scenario of no sand policy.

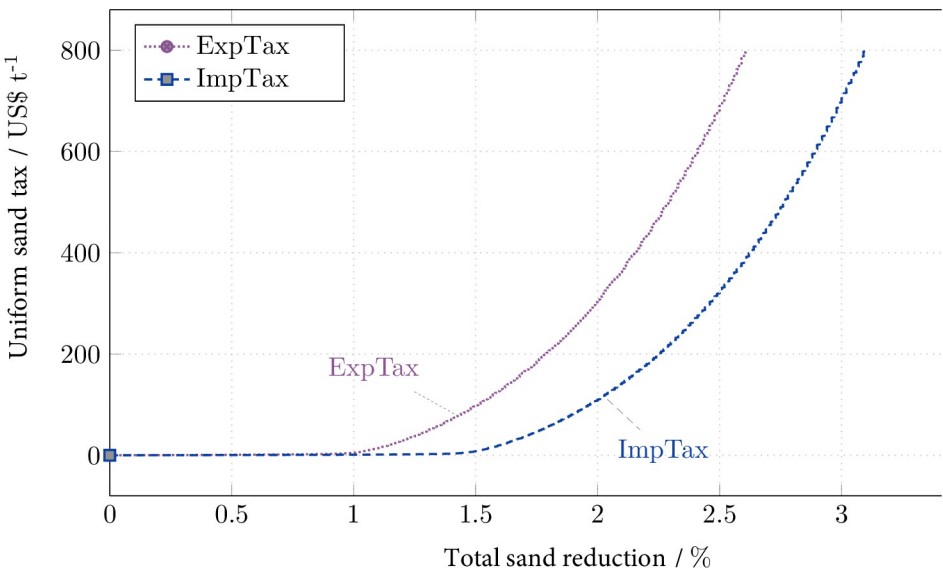

**Fig 2. Tax rate (marginal reduction cost) as a function of the total reduction of sand extraction under the export/ import tax (ExpTax/ImpTax).**

The use of the output tax in all exporting countries (OutTax) results in a positive, convex and increasing welfare effect as a function of the corresponding sand reduction for all exporters. (The extension of the solution scope would probably lead to a welfare maximum, the optimal tax, and a consecutive decline in the positive welfare effects. Because of the relatively low sand demand elasticity, the scope of feasible model solutions encompasses the rising branch before the maximum only.) Whereas the total sand extraction can be reduced by approximately 70%, the achievable welfare gains vary from 0.06% in Malaysia and Myanmar to over 0.3% in the Philippines and Vietnam, and up to 1.4% in Cambodia. Singapore's corresponding welfare effect mirrors those of the exporters: the welfare effect as a function of the sand reduction is negative, concave and decreasing; the magnitude of the welfare loss reaches 0.08%.

## 8. Discussion, robustness check and sensitivity analysis

This section provides a critical discussion of the model and the simulations considering limitations and uncertainty. The most characteristic feature of the analyzed model is the detailed, theory-based representation of international trade. The model structure and the simulations draw on benchmark data from official statistical bureaus describing the current economic environment. Thus, the model is tailored to studying the effects of taxation on international trade given the current trade patterns.

If scholars or policymakers intend to study long-term developments and policies, a *Hotelling* type model of intertemporally optimal resource extraction with a limited, nonrenewable sand deposit can be the preferable model type. Our analysis has a short- to medium-term perspective. It assumes that sand deposits are (at least to some extent) renewable, replaceable or available in sufficient quantities within the examined time horizon. A long-term analysis would be complicated by uncertainty of technical progress, particularly, the development of substitutes for conventional sand, which might substantially reduce the need for sand in construction or even make it obsolete.

The simulation results have revealed substantial differences between the effects of trade policy and output taxation. These differences hinge on the implicit possibility of shifting sand sales in the model from the export to the domestic market. If this possibility becomes more restricted and more of the extracted sand is shipped to Singapore, the effects of trade policy will be closer to those of output taxation, i.e., the trade policy will become more effective with respect to the reduction of sand extraction.

In our model, the possibility to replace different production factors, different (domestic and imported) goods or exports and domestic sales by each other in production and consumption hinges on the stylized assumption and calibration of constant elasticity of substitution (CES) functions (for details see the supplementary part II S1 Appendix, particularly sections 2, 3 and 5). The use of such CES functions is standard in computational economic modelling (and common in economic theory), because it represents substitution possibilities in a straight-forward, constant and theory-consistent way, it enables clear-cut, closed algebraic solutions, the design of different functional behavior types based on the same CES concept, a straight-forward numerical calibration and fast, unique numerical solutions. For example, the CES consumption function used in our model (S1 Fig A1 in S1 Appendix) implies that a representative consumer will adjust the composition of her consumption bundle when the (relative) prices for goods change (induced by policies).

Unofficial sand trade is not captured by the data. Therefore, we have performed a robustness check with extended sand demand and hence trade. S1 Figs R1 to R3 in S1 Appendix in the supplementary part I illustrate the corresponding simulation results for the HigDem scenario with the assumption that Singapore's sand demand will increase fivefold compared to

the previous standard policy scenarios. The robustness check has demonstrated that future changes in Singapore's sand demand can create greater than proportional changes in welfare effects for both Singapore as the sand importer, and sand exporters.

In particular, according to this robustness check, the welfare effects of the export tax (ExpTax/HigDem) that are negative for the importer Singapore but positive for the exporters, rise by almost an order of magnitude (except in the Philippines) compared to the standard scenario (ExpTax). The corresponding maximum achievable total sand reduction more than doubles to 7.5% under ExpTax/HigDem and 8.0% under ImpTax/HigDem (S1 Fig R1 b in S1 Appendix compared with Fig 2 above). The welfare gain that Singapore can achieve via the import tax (ImpTax/HigDem) remains the same as before (3% under ImpTax). Most welfare effects and the corresponding total sand reductions induced by the output tax (OutTax/Hig-Dem), in contrast, hardly change compared to the respective values under the standard scenarios, except in Singapore and Myanmar, where welfare effects rise by almost an order of magnitude compared to the effects under the standard scenario (OutTax).

On the one hand, no precise official forecasts of Singapore's sand demand are available. On the other hand, any forecast of sand demand, supply and trade is subject to uncertainty of future economic growth and technical progress that might introduce alternative less sand-intensive production techniques and thus reduce sand demand. Therefore, the prediction of future sand demand is the primary source of uncertainty. Nonetheless, the advantages of pricing sand extraction (OutTax), implemented either as a tax or via the *Sand Extraction Allowances Trading Scheme (SEATS)*, are qualitatively robust to the robustness check.

To assess the uncertainty in key model parameter values, we perform a *sensitivity analysis*. The results are shown in S1 Figs S1 to S3 of the supplementary part I in S1 Appendix. First, we lower and raise the parameter values governing the trade elasticity in all sectors by one standard deviation based on the estimates of Caliendo and Parro [18]. The effect on sand extraction and welfare is close to zero under the output tax (OutTax) and is small under the export tax (ExpTax). Second, we change the elasticity of substitution between intermediate inputs in the construction (*CONS*) and nonmetallic minerals (*NMMS*) sectors from 0 to 0.25 and 0.75, which varies the substitution possibilities between inputs within a moderate range. (These two sectors absorb most of the sand supply. The value of zero implies that there are no substitution possibilities between different inputs.) The effect on sand extraction and welfare is significant. For instance, the achievable sand reduction via the export tax doubles, and Singapore's welfare loss created by the output tax decreases by 50% if 0.75 is assumed. Although these effects are dominated by the uncertainty of the future sand demand discussed above, alterations of elasticities can increase or decrease the differential between the effects of trade policy and output taxation.

Another unknown factor is the socially optimal rate of the sand tax, i.e., the correct rate of the *Pigou* tax that reflects the total social (especially environmental) marginal damage. Because this information is unavailable to date, this analysis has exploited a wide range of feasible tax rates. From a modeling perspective, feasibility refers to the solution space of well-defined, unique market equilibria. From a policy perspective, feasibility refers to reasonable tax rates as depicted by Fig 1 above. Such tax rates reach nearly US-$20 per ton under the output tax (Out-Tax) scenario. Given sand prices have been between approximately US-$4 and 9 per ton during the preceding ten years [3, 24], the price of US-$20 per ton implies very substantial taxation. Tax rates reaching hundreds of dollars, as examined in the trade policy scenarios ExpTax and ImpTax with export and import taxes, appear to be prohibitively high and politically unrealistic. Against this backdrop, the range of tax rates examined in this analysis seems to be sufficient with regard to practical policy implications.

## 9. Policy solutions for sand extraction and trade

The policy discussion and model-based analysis provide the following key insight:

*The policy simulation results are in favor of a uniform output tax imposed on the total sand extraction in all sand extracting countries.*

Furthermore, by studying and interpreting the results in detail, a number of subtly nuanced insights are gained. The output tax (OutTax) can achieve a wide range of reductions of sand extraction at moderate marginal costs represented by moderate tax rates. While all sand exporters gain from higher gross sand prices (including the tax) and resulting tax revenues, Singapore's welfare losses are minor. This policy goes beyond Singapore's responsibility for the externalities of its sand consumption, because such taxation significantly affects *domestic* sand sales of the Southeast Asian sand extractors. The output tax can equivalently be implemented as a *Sand Extraction Allowances Trading Scheme (SEATS)*. In either case, the implementation poses challenges. In particular, it requires coordinated action by the exporters to implement it and capture the entire sand extraction market.

If only a part of the sand market is taxed in the trade policy scenarios (ExpTax or ImpTax), the untaxed part of the market will absorb most of the excess supply. This effect renders trade policies rather ineffective for reducing sand extraction. Although not explicitly visible in the model, the untaxed market may include illegal sand sales and exports that undermine any taxation or regulation of legal sand extraction and trade. Thus, full coverage and strict monitoring of sand taxation in all exporting countries are inevitable. Satellite imagery can support monitoring activities [25].

The export tax (ExpTax), on the contrary, has the advantage of benefiting the poorest exporters, Cambodia and to a lesser extent Myanmar. The other exporters, however, are slightly worse off than without a sand policy. A robustness check (HigDem) indicates that Singapore's expected increasing future sand demand can enlarge the welfare effects of the export tax by an order of magnitude.

The primary advantage of the import tax (ImpTax) policy is the opportunity for Singapore to implement it unilaterally following its green policy principles without the requirement of international coordination. In this vein, the import tax can act as a temporary compromise as long as a coordinated policy covering the entire sand extraction cannot be achieved. Nonetheless, policymakers should engage in international negotiations to capture the social and environmental benefits of the output tax. The potential welfare gains should create the right economic incentive for the Southeast Asian exporters, while the "green front-runner" role should create an incentive for Singapore or any other modern urban area.

The existing export bans do not generate official revenues but instead encourage smuggling and unofficial rent-seeking. The introduction of taxes (tariffs), on the contrary, would generate official state revenues that could be used for social or environmental purposes related to sand mining, particularly as a compensation to those whose welfare is reduced by sand mining.

Finally, worldwide, cement production is a significant source of $CO_2$ (from process emissions and energy use for heating) [26] and hence contributes to the climate change externality. The taxation of sand is certainly not the appropriate first-best instrument to address this externality. Nonetheless, in the absence of $CO_2$ pricing, sand taxation will mitigate climate change by making cement production costlier as a co-benefit. Therefore, as long as neither a technological solution to avoid the release of $CO_2$ emissions [26] nor a global climate policy solution, covering countries or regions such as Singapore and Southeast Asia, is in sight, climate change mitigation is another argument in favor of sand taxation.

## 10. Concluding remarks and research outlook

This article has introduced a unique advanced trade model within a complex global general equilibrium framework. As a central novel feature, it represents sand extraction and trade explicitly. Besides, it covers 15 production sectors and 16 countries or world regions and their complex economic linkages. The model was used to analyze novel policy strategies regulating the extraction, exports or imports of sand. Taking Singapore and its Southeast Asian neighbor countries as a prominent example, it has derived policy recommendations with global relevance.

The analysis of a general equilibrium model allows this study to consider the connections between the sectors in each economy. This is particularly important for raw materials, such as sand, that reach consumers through complex value chains rather than directly. The trade theoretical foundation of the model, which rests on a Ricardian (productivity-based) comparative advantage, is particularly plausible for primary sectors.

Although this study will not resolve the global problem of harmful sand extraction and limited sand reserves, it has demonstrated that smart policies can, at least to some extent, reconcile "green growth" with sand imports if such policies are implemented and controlled rigorously. Independent of local particularities and corresponding parameter values in the model, the qualitative policy insights from the presented case study are applicable to other expanding (mega) cities, urban areas and growing economies in the world. Particularly, we find the following implications for the operational policy implementation in developing countries acting as sand exporters and industrialized countries acting as sand importers:

First, the developing countries may introduce a coordinated uniform output tax on preferably all sand extraction activities to avoid any "sand leakage" to other untaxed countries or sectors. As a result, they will not only benefit from improved environmental and social conditions due to less sand mining, but probably also from welfare gains due to the increased sand price (including the tax rate). These potential benefits may encourage their governments to engage in the necessary coordination of the tax (rate) across borders. Coordination failure will result in efficiency losses and hence smaller welfare gains. In any case, according to the model simulations, losses for the industrialized countries are likely to be relatively small. A progressive redistribution scheme could avoid an undesirable burden for low-income households in these developing countries.

Second, following the same rationale, the developing countries may introduce export tariffs on sand (with or without coordination among the developing countries). Because this policy is restricted to the exported part of the extracted sand, it will be less effective than the output tax in reducing sand extraction, and the welfare effects for the developing countries will be more diverse and possibly negative for some exporters.

Third, the main advantage of the sand import tax is the opportunity for the industrialized countries to introduce it without coordinating it with the developing countries (with or without coordination among the industrialized countries) to mitigate sand consumption and to follow an exemplary "green front-runner" strategy. This advantage is, however, concomitant with expected small welfare losses of the developing countries, because the tax revenues accrue to the industrialized countries.

Fourth, by making sand more expensive, the derived policies are expected to spur technological progress searching for possibilities of concrete recycling or the replacement of natural sand by modified sand from desserts or other man-made (recycled) or natural materials and to make such alternatives to sand competitive. (The fine desert sand is grinded and pressed into pellets [27]).

In this way, it is hoped that this study has pointed to a promising avenue for further conceptual research and case studies in the domain of public policy and among different scientific disciplines. For example, natural scientists may search for more precise information on the damage of sand mining. Economic modelers may evaluate the proposed policies for other countries (or cities) and think about the design of further sand mining and trade policies. Finally, economists and political scientists may explore approaches to the coordinated (sand tax) policy implementation.

The further political neglect of the global sand challenge, however, could severely hinder urban economic development in terms of housing and infrastructure, which is not only crucial for developing and emerging countries, but also for industrialized countries.

## Supporting information

**S1 Appendix. Supplementary online figures and appendix.**
(PDF)

## Acknowledgments

We thank Ulrike Grote for her support.

## Author Contributions

**Conceptualization:** Michael Hübler.

**Data curation:** Frank Pothen.

**Formal analysis:** Michael Hübler, Frank Pothen.

**Funding acquisition:** Michael Hübler, Frank Pothen.

**Methodology:** Michael Hübler, Frank Pothen.

**Project administration:** Michael Hübler, Frank Pothen.

**Software:** Frank Pothen.

**Validation:** Michael Hübler, Frank Pothen.

**Visualization:** Frank Pothen.

**Writing – original draft:** Michael Hübler.

**Writing – review & editing:** Michael Hübler.

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
