## [Decision Letter · Decision Letter 0]

4 Nov 2020

PONE-D-20-26007

Can Smart Policies Solve the Sand Mining Problem?

PLOS ONE

Dear Dr. Hübler,

Thank you for submitting your manuscript to PLOS ONE. After careful consideration, we feel that it has merit but does not fully meet PLOS ONE’s publication criteria as it currently stands. Therefore, we invite you to submit a revised version of the manuscript that addresses the points raised during the review process.

As mentioned by both reviewers, the paper requires minor issues to be revised and corrected. Below you can find the reviewers' comments.

We look forward to receiving your revised manuscript.

Kind regards,

Eda Ustaoglu, PhD

Academic Editor

PLOS ONE

Journal Requirements:

Reviewers' comments:

Reviewer's Responses to Questions

**Comments to the Author**

1. Is the manuscript technically sound, and do the data support the conclusions?

Reviewer #1: Yes

Reviewer #2: Yes

2. Has the statistical analysis been performed appropriately and rigorously? 

Reviewer #1: Yes

Reviewer #2: Yes

3. Have the authors made all data underlying the findings in their manuscript fully available?

Reviewer #1: Yes

Reviewer #2: No

4. Is the manuscript presented in an intelligible fashion and written in standard English?

Reviewer #1: Yes

Reviewer #2: Yes

5. Review Comments to the Author

Reviewer #1: The subject of this manuscript is very practical.

The smart policies could be applied in developed countries and make a plane for developing countries.

In the discussion section, it seems better to consider operational strategies for third world and developed countries separately.

Reviewer #2: The paper makes a commendable attempt to address the issue of exploitative sand mining. They focus primarily on sand flows to Singapore from South East Asia. As such I believe that the paper is of a very high standard and is motivated by both rigorous theory and data. I would like to suggest some minor changes in this paper to deem it fit for publication at PLOS One, in my opinion:

1) I think the conclusion section needs to be rewritten to reflect the salient features of the model and its conclusions. It reads like a standard "need for futher research" paragraph and I dont think the authors put the best foot forward on the novelty of the model that they developed.

2) The theoretical model uses a CES function to model demand. While I understand this may be important for computational purposes, I think it best to say that this is an assumption and perhaps explain how they justified using this function.

3) Equation 2 in the appendix seems incomeplete to me. Why is there no error term?

6. PLOS authors have the option to publish the peer review history of their article (what does this mean?). If published, this will include your full peer review and any attached files.

Reviewer #1: No

Reviewer #2: **Yes: **Tamanna Adhikari

---

## [Author Response · Author response to Decision Letter 0]

13 Jan 2021

5. Review Comments to the Author

Reviewer #1: The subject of this manuscript is very practical.

The smart policies could be applied in developed countries and make a plane for developing countries.

In the discussion section, it seems better to consider operational strategies for third world and developed countries separately.

Reply: Thank you very much for your very helpful comment that helped us improve the practical policy relevance of our study.

Following your recommendation, we have summarized operational policy strategies referring to developing and industrialized countries and positioned them prominently in the center of the conclusion (pp. 17-18). This accords with Reviewer #2 who finds that the conclusion should highlight the novelty of our modelling work:

“Particularly, we find the following implications for the operational policy implementation in developing countries acting as sand exporters and industrialized countries acting as sand importers:

First, the developing countries may introduce a coordinated uniform output tax on preferably all sand extraction activities to avoid any “sand leakage” to other untaxed countries or sectors. As a result, they will not only benefit from improved environmental and social conditions due to less sand mining, but probably also from welfare gains due to the increased sand price (including the tax rate). These potential benefits may encourage their governments to engage in the necessary coordination of the tax (rate) across borders. Coordination failure will result in efficiency losses and hence smaller welfare gains. In any case, according to the model simulations, losses for the industrialized countries are likely to be relatively small. A progressive redistribution scheme could avoid an undesirable burden for low-income households in these developing countries.

Second, following the same rationale, the developing countries may introduce export tariffs on sand (with or without coordination among the developing countries). Because this policy is restricted to the exported part of the extracted sand, it will be less effective than the output tax in reducing sand extraction, and the welfare effects for the developing countries will be more diverse and possibly negative for some exporters.

Third, the main advantage of the sand import tax is the opportunity for the industrialized countries to introduce it without coordinating it with the developing countries (with or without coordination among the industrialized countries) to mitigate sand consumption and to follow an exemplary “green front-runner” strategy. This advantage is, however, concomitant with expected small welfare losses of the developing countries, because the tax revenues accrue to the industrialized countries.

Fourth, by making sand more expensive, the derived policies are expected to spur technological progress searching for possibilities of concrete recycling or the replacement of natural sand by modified sand from desserts or other man-made (recycled) or natural materials and to make such alternatives to sand competitive.” 

Furthermore, we have added a new reference to a technological solution for avoiding the release of CO2 emissions in cement production, published in PNAS in 2020 at the end of the policy solutions in section 9 (p. 16):

“Finally, worldwide, cement production is a significant source of CO2 (from process emissions and energy use for heating, Ellis et al., 2020)… Therefore, as long as neither a technological solution to avoid the release of CO2 emissions (Ellis et al., 2020) nor a global climate policy solution…”

We hope, these paragraphs underline the policy relevance for developing and industrialized countries. If these strategies should be extended and/or positioned elsewhere, please let us know.

Reviewer #2: The paper makes a commendable attempt to address the issue of 

exploitative sand mining. They focus primarily on sand flows to Singapore from South East Asia. As such I believe that the paper is of a very high standard and is motivated by both rigorous theory and data. I would like to suggest some minor changes in this paper to deem it fit for publication at PLOS One, in my opinion:

Reply: Thank you very much for your very thoughtful comments that helped us improve the manuscript.

1) I think the conclusion section needs to be rewritten to reflect the salient features of the model and its conclusions. It reads like a standard "need for futher research" paragraph and I dont think the authors put the best foot forward on the novelty of the model that they developed.

Reply: Thanks to his helpful comment, we have included the following paragraphs to the beginning of the conclusion (p. 16): 

“This article has introduced a unique advanced trade model within a complex global general equilibrium framework. As a central novel feature, it represents sand extraction and trade explicitly. Besides, it covers 15 production sectors and 16 countries or world regions and their complex economic linkages. The model was used to analyze novel policy strategies regulating the extraction, exports or imports of sand. Taking Singapore and its Southeast Asian neighbor countries as a prominent example, it has derived policy recommendations it has derived policy recommendations with global relevance.

The analysis of a general equilibrium model allows this study to consider the connections between the sectors in each economy. This is particularly important for raw materials, such as sand, that reach consumers through complex value chains rather than directly. The trade theoretical foundation of the model, which rests on a Ricardian (productivity-based) comparative advantage, is particularly plausible for primary sectors.”

Following Reviewer #1’s recommendation, we have summarized operational policy strategies for developing and industrialized countries and positioned them in the center of the conclusion, which also underlines the novelty and importance of our model analysis (p. 17).

Additionally, we have added the following remark to the policy recommendations in the concluding section (pp. 17, 18):

“Fourth, by making sand more expensive, the derived policies are expected to spur technological progress searching for possibilities of concrete recycling or the replacement of natural sand by modified sand from desserts14 or other man-made (recycled) or natural materials and to make such alternatives to sand competitive. >>Footnote 14: The fine desert sand is grinded and pressed into pellets (Gassmann, 2019).”

As explained above, we have added a new reference to a technological solution for avoiding the release of CO2 emissions in cement production, published in PNAS in 2020 at the end of the policy solutions in section 9 (p. 16):

“Finally, worldwide, cement production is a significant source of CO2 (from process emissions and energy use for heating, Ellis et al., 2020)… Therefore, as long as neither a technological solution to avoid the release of CO2 emissions (Ellis et al., 2020) nor a global climate policy solution…”

2) The theoretical model uses a CES function to model demand. While I understand this may be important for computational purposes, I think it best to say that this is an assumption and perhaps explain how they justified using this function.

Reply: Thank you for this hint; we agree that the main text should critically mention the use of CES functions. Accordingly, we have added the following discussion to section 8 (p. 13, 2nd par.):

“In our model, the possibility to replace different production factors, different (domestic and imported) goods or exports and domestic sales by each other in production and consumption hinges on the stylized assumption and calibration of constant elasticity of substitution (CES) functions (for details see the supplementary part II Appendix, particularly sections 2, 3 and 5). The use of such CES functions is standard in computational economic modelling (and common in economic theory), because it represents substitution possibilities in a straight-forward, constant and theory-consistent way, it enables clear-cut, closed algebraic solutions, the design of different functional behavior types based on the same CES concept, a straight-forward numerical calibration and fast, unique numerical solutions. For example, the CES consumption function used in our model (S1 Fig A1 in the appendix) implies that a representative consumer will adjust the composition of her consumption bundle when the (relative) prices for goods change (induced by policies).”

3) Equation 2 in the appendix seems incomeplete to me. Why is there no error term?

Reply: This is a valid question. If Eq. 2 were standing alone, it would indeed be required. Eq. 2, however, is plugged into Eq. 1, containing an error, replacing the corresponding “log delta” term there. Then the resulting equation is estimated. Hence, an error term in Eq. 2 is superfluous. We have clarified this in the appendix by writing above Eq. 2 (p. 16): “They are approximated by equation (2), which is plugged into equation (1).

In summary, we are grateful for these comments and would like to implement any further suggestions for improvements.

---

## [Editor Report · Decision Letter 1]

24 Feb 2021

PONE-D-20-26007R1

Can Smart Policies Solve the Sand Mining Problem?

PLOS ONE

Dear Dr. Hübler,

Thank you for submitting your manuscript to PLOS ONE. After careful consideration, we feel that it has merit but does not fully meet PLOS ONE’s publication criteria as it currently stands. Therefore, we invite you to submit a revised version of the manuscript that addresses the points raised during the review process.

There are only minor issues regarding the revisions in the citing of the references. Please re-arrange the references accordingly.

We look forward to receiving your revised manuscript.

Kind regards,

Eda Ustaoglu, PhD

Academic Editor

PLOS ONE

Journal Requirements:

Additional Editor Comments (if provided):

Please re-organise the references in cases where it is better to highlight the name(s) of the references in the text for ex. "Hoogmartens et al. (2014) [15] examine sand extraction in Flanders based on a..." Here it is better to keep 'Hoogmartens et al. [15]' but delete only the year "(2014)" . But if the references are given at the end of the sentences in pharanthesis e.g. "...sand mining as a major contributor to the geomorphic changes (Nguyen et al., 2015; Darby et al, 2016)" in this case replace these references with the corresponding numbers only i.e. "[13,14]"

---

## [Author Response · Author response to Decision Letter 1]

1 Mar 2021

Dear Editor Prof. Eda Ustaoglu,

Thank you very much for the opportunity to finalize our manuscript. 

In the previous revision of our manuscript, we had added a couple of references (Ellis et al., 2020; Gassmann, 2019) that we cited in the reply letter. Furthermore, we ensured that all references that appear in the text also appear consistently in the reference list and that there are not any superfluous references included in the list. We rearranged the order of the references in the list according to the journal guidelines. In the current revision, we checked this aspect and believe that the reference list is complete and correct.

In the last revision, we also replaced the author names and years appearing in the text by consecutive numbers […]. In the current revision, we reinserted the author name before […] where references appear as a subject or object of a sentence as you suggested (pp. 2, 5, 7, 10, 14; see the revised manuscript with track changes).

Please let us know, if further improvements are recommendable or required. We would appreciate and like to implement any further suggestion.

Sincerely yours,

The authors

---

## [Editor Report · Decision Letter 2]

8 Mar 2021

Can Smart Policies Solve the Sand Mining Problem?

PONE-D-20-26007R2

Dear Dr. Hübler,

We’re pleased to inform you that your manuscript has been judged scientifically suitable for publication and will be formally accepted for publication once it meets all outstanding technical requirements.

Kind regards,

Eda Ustaoglu, PhD

Academic Editor

PLOS ONE
---

## [Editor Report · Acceptance letter]

26 Mar 2021

PONE-D-20-26007R2 

Can smart policies solve the sand mining problem? 

Dear Dr. Hübler:

I'm pleased to inform you that your manuscript has been deemed suitable for publication in PLOS ONE. Congratulations! Your manuscript is now with our production department. 

Kind regards, 

on behalf of

Dr. Eda Ustaoglu 

Academic Editor

PLOS ONE